# ANALYZING THE PERFORMANCE OF VERTICAL WIND PROFILERS IN RAIN EVENTS

Adriel J. Carvalho[1], Francisco A. Leite Neto[2], and Denisson Q. Oliveira[1]

[1]Institute of Electrical Energy, Federal University of Maranhão, São Luís, Brazil
[2]Applied Meteorology Laboratory, Geosciences Institute, Federal University of Rio de Janeiro, Rio de Janeiro, Brazil

**Correspondence:** Denisson Oliveira (dq.oliveira@ufma.br)

**Abstract.** This paper quantitatively analyzes the performance of SODAR and LIDAR wind profilers during precipitation events, focusing on their Range Availability (RA) and the representativeness of wind measurements. The wind profile and supporting meteorological data have been collected in Barreirinhas and Paulino Neves, Maranhão, Brazil, at various locations, both near and far from the shoreline. The results show that precipitation affects the RA of SODAR, which, although it recovers quickly after the rain, shows significant drops in more consistent events. On the other hand, the LIDAR near the coast had little influence from rainfall on its RA. In contrast, when the LIDAR is far from the coast, it showed more significant variability, with drops in RA not necessarily linked to rainfall events. The investigation has concluded that the location and specific meteorological conditions significantly influence the performance of these wind profilers and should be considered when choosing the technology for estimating the vertical wind profile.

**KEYWORDS:** Wind profilers, LIDAR, SODAR, Wind Speed, Rainfall

## 1 Introduction

In the last decade, wind generation has grown significantly worldwide. This growth aligns with the global trend of increasing the share of clean sources in the electricity matrix, one of the cores of sustainable development based on low-carbon solutions. Wind power has consistently increased its share, surpassing the milestone of 1 TW of installed power worldwide in 2023 (Council, Global Wind Energy, 2024).

The large number of projects granted licenses and the increase in the height of wind turbines, as well as the need to optimize generation costs, has led to research into better understanding and predicting the phenomena that impact wind power generation. Although countries have different legislation, in all cases, governments and regulatory agencies require a period of uninterrupted measurements of wind speed and direction at a given site to approve projects. Such requirements make it vital to develop methods and technologies capable of adequately estimating wind resources reliably at heights that are difficult to reach with anemometric towers.

Vertical wind profilers, such as SODAR and LIDAR, work based on the Doppler effect and can reliably estimate wind speed and direction at different height intervals, previously configured by the user, and with a more extensive range than conventional

instruments. These sensors vary in their performance, especially in the face of temperature inhomogeneities in the atmosphere
and noise, in the case of SODAR, and the amount of aerosols in the case of LIDAR.

Wind profilers are versatile, reliable, robust, and reusable solutions that reduce the costs of measurement campaigns and allow for a more accurate characterization of the wind resource available at a given site. Unlike conventional measurements using anemometric towers, profilers do not require approximations or extrapolations to determine wind speed and direction at greater heights. Such a feature makes them more attractive for measurement campaigns, as the heights observed are consistent with the trend for the hub height of wind turbines to rise. This increase in the hub height allows the exploration of wind resources at higher heights but also requires instruments that are equally capable of observing the variables at these heights.

## 1.1   State of the Art and Contributions

With the development of remote profilers, several previous investigations have already carried out short and long-term comparisons between conventional measurers (mechanical and sonic anemometers), radiosondes and remote profilers. The published results have already shown that the measurements correlate well, although with some peculiarities linked to the operation of conventional instruments compared to remote profilers, validating the use of profilers in remote wind sensing for power generation and other aviation-related purposes.

Frehlich et al. (2008) investigated the stable boundary layer in suburban areas, focusing on small-scale turbulence, comparing the Tethered Lifting System (TLS) and a Doppler LIDAR, and developing a processing algorithm for assessing turbulence statistics. Kumer et al. (2014) compared the data acquired from scanning LIDAR, a vertical LIDAR, and radiosondes. The results showed that the measurements correlate better as the height increases. D. Kim et al. (2016) compared ground-based LIDAR and met mast measurements over various terrain conditions, showing good measurement reliability. Dubov et al. (2017) compare wind data measurements of SODAR and meteorological mast at different height levels on flat terrain, showing good correlations (0.943) over the period. Dubov et al. (2018) compare LIDAR and a meteorological mast, showing a good correlation between all heights. Other similar investigations have been performed by Chaurasiya et al. (2017) and Khan and Tariq (2018), finding high correlations between measurements.

Other comparisons between wind profilers and other technologies are available in the literature. Zhou and Bu (2021) compared measures from a LIDAR with L-band sounding radar and wind cup. J. Y. He et al. (2022) compared measures from SODAR to a microwave radiometer. Sinha et al. (2018) investigated the application of multiparameter cost function to measurements from SODAR and radar, indicating good performance and complementarity between both wind profiling systems. Buzdugan and Stefan (2020) compared LIDAR and SODAR measures to aircraft observations, while Buzdugan et al. (2021) compared wind profilers to surveillance radars. Finn et al. (2017) compared the wind profile from SODAR to the measures from unmanned aerial vehicle-based tomography, finding similar levels of correspondence.

Lang and McKeogh (2011) compared measurements from LIDAR and SODAR profilers with measurements from a meteorological tower instrumented with cup anemometers in a typical semi-complex upland terrain. The results showed a good correlation between measurements from conventional instruments and wind profilers but also indicated that wind profilers perform better for estimating winds at low speeds. Kelley et al. (2007) compared LIDAR and SODAR profilers and sonic

anemometers installed on a 116 m tower. The results showed a good correlation between the measurements, especially between the LIDAR and sonic anemometers.

Torres Junior et al. (2022) investigated and compared the performance of SODAR and LIDAR profilers operating simultaneously over a short period at two points: one in an urban area and the other at a point near the coast. The results showed that the performances of both devices are similar, with a good correlation in wind measurements, although LIDAR performed better near the coast. Liu et al. (2019) reviewed advances in LIDAR technology and its different applications. Gao et al. (2022) presented a technique to describe the 3-D wind field in complex terrain more appropriately using a single LIDAR in conjunction with the Taylor series and Ridge-DI method.

Wolz et al. (2024) compared wind measurements from a triple Doppler LIDAR virtual tower configuration with those from a sonic anemometer located at 90 m height on an instrumented tower and with those from two single Doppler LIDARs to assess the effect of the horizontal homogeneity assumption used for single Doppler LIDAR applications on measurement accuracy. The results showed that a single LIDAR provides reliable wind speed and direction measurements over heterogeneous but flat terrain in different scan configurations.

Although the literature presents comparisons between wind profiler measurements in experiments with different time durations and located in regions of different latitudes, the authors did not find any information about the performance of these sensors in specific meteorological conditions, such as rain or comparisons made during measurement campaigns in regions with high rainfall. Rain is an important event from the perspective of wind profilers because it can modify the atmosphere and influence the operation of the sensors. In the case of LIDAR, the particles and aerosols that reflect the signal are "washed out" of the atmosphere, resulting in a lack of reflective particles for a period. Such conditions cause a drop in LIDAR efficiency until these aerosols recover. For SODAR, on the other hand, rain also diminishes atmospheric inhomogeneities by standardizing the temperature over a wide range of heights and reducing temperature gradients between different layers, influencing the reflection of sound waves. In addition to this, there is also rain noise, which can cause a loss of sensitivity in SODAR measurements.

Investigating and understanding the operation of these profilers and the influence of rainfall conditions on their performance is essential for the wind industry. As such, a drop in their performance can impact the quality and quantity of the data observed, deteriorating the rates at which valid data is obtained. A reduction in the data collected can impact the validation of measurement campaigns by certifying companies to obtain permits and deteriorate the quality of the annual energy generated forecast, increasing the financial risks of the wind project.

This paper aims to contribute by analyzing the observations made with SODAR and LIDAR remote profilers to investigate their performance when inserted into precipitation events in different locations ranging from hundreds of meters to tens of kilometers from the coast. The aim of this analysis is not to pinpoint a superior technology between the two wind profilers analyzed but to investigate their performance in everyday situations for the wind energy industry and to determine the influences and how soon the equipment returns to normal wind profiling conditions in each situation during and after rainfall events of different intensities and durations. Such quantitative comparisons between the two technologies do not intend to speculate results in different climates or geographic conditions due to numerous variables that can affect the performance of the sensors that could not be quantified, such as high roughness length, low humidity, and temperature. However, the methodology presented here is

general and could be replicated in other regions if data are available. This investigation is carried out with the support of 14 months of observations in a region with well-defined dry and rainy seasons of approximately equal duration.

## 2 Methodology

The data used in this investigation have been obtained in Barreirinhas and Paulino Neves, Maranhão, Brazil. This region is inside the Brazilian equatorial margin, with a large area with great wind potential. This region of the Brazilian coast has a high availability of wind resources, mainly due to the trade winds along the equatorial coast. Previous investigations, such as in Assireu et al. (2022) and Pimenta et al. (2023), have preliminarily described atmospheric flows in the equatorial region, pointing out physical processes that modify the structure of the atmospheric boundary layer, impacting wind speed, direction, vertical wind shear, and turbulence.

### 2.1 Data Acquisition

This investigation uses data from various sensors. The vertical wind profile (horizontal speed and direction) has been analyzed using Leosphere's LIDAR Windcube V2 and Scintec's SODAR MFAS. Observations have been made at 20 different heights, ranging from 40 to 200 m, at 10 m intervals, and at heights of 220 m, 240 m, and 260 m, with 10 min averages.

In addition to the vertical wind profile, a micrometeorological station installed next to the two wind profilers has collected meteorological variables throughout the campaign. A weighbridge rain gauge, accurate to 0.1 mm, measured precipitation, and the data were processed in accumulations of 1 min. A 2D sonic anemometer recorded wind speed at 10 m above the ground. A thermometer installed 3 m above the ground monitored temperature.

### 2.2 Sensor location and measurement range

During the measurement period, the LIDAR operated at various points, as described in 1, to detect seasonal variations in wind resources along the coast.

Sodar operated at point P1, described in Table 1, from September 16th, 2021, to July 27th, 2022, representing 312 days. Due to the different locations where the LIDAR was placed, the analysis of its data was separated into two sets: (1) close to the coast, covering points P0, P1, P2, P3, P6, and P7, and (2) far from the coast, covering points P4 and P5, from Table 1.

**Table 1.** Geographical points and LIDAR operating ranges.

| Period | Point | Days | Location | Distance to the shoreline |
|---|---|---|---|---|
| SEP 16th, 2021-NOV 9th, 2021 | P1 | 54 | 2°43'29.6"S 42°34'30.7"W | 5 km |
| NOV 12th, 2021-DEC 15th, 2021 | P0 | 33 | 2°41'38.8"S 42°33'17.3"W | 1.6 km |
| DEC 15th, 2021-JAN 27th, 2022 | P2 | 44 | 2°43'30.6"S 42°36'23.4"W | 7.8 km |
| JAN 28th, 2022-APR 18th, 2022 | P3 | 80 | 2°44'00.7"S 42°35'22.3"W | 7.1 km |
| APR 20th, 2022-JUN 13th, 2022 | P4 | 54 | 2°45'32.5"S 42°48'25.7"W | 24 km |
| JUN 15th, 2022-JUL 27th, 2022 | P5 | 42 | 2°47'14.5"S 42°51'20.6"W | 30 km |
| JUL 29th, 2022-SEP 15th, 2022 | P6 | 48 | 2°44'00.7"S 42°35'22.3"W | 7 km |
| SEP 16th, 2022- NOV 08th, 2022 | P7 | 53 | 2°43'30.6"S 42°36'23.4"W | 7.8 km |

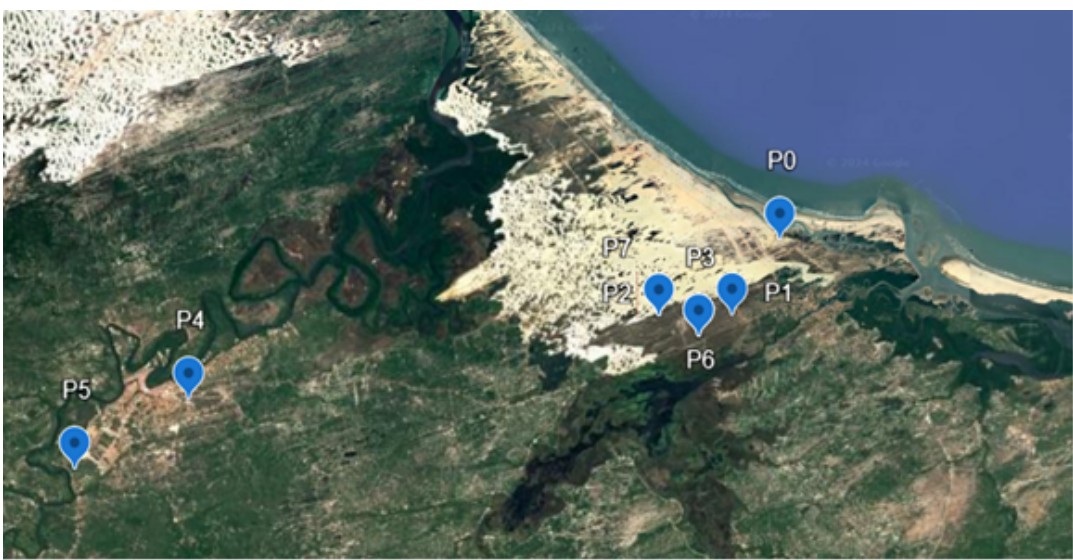

**Figure 1.** Geographical representation of points P0 to P7 in the map.

## 2.3 Data Analysis Criteria

The data analysis followed the criteria described below.

### 2.3.1 Range Availability (RA)

It is the ratio between the heights at which wind observations were made in each reading and the total number of heights defined for the sensor's vertical profile.

$$RA = \frac{\text{amount of observed heights}}{20} \tag{1}$$

### 2.3.2 Maximum Range ($R_{max}$)

It represents the longest range the equipment could estimate the wind speed. The maximum range is a valuable metric to indicate the equipment's loss of range, as losses tend to occur initially at the highest altitudes. However, in some events, the lost observations occurred at intermediate altitudes, not affecting the maximum range. To address these wide-ranging losses, RA has been used, which encompasses all the observations made by the equipment.

### 2.3.3 Rain Events

Events were characterized as precipitation accumulation occurring in a 10-minute interval (starting with the full hour). Each event has the following parameters:

- **Cumulative (C10):** sum of the 1 min rainfall accumulations that occurred within the 10 min interval;

- **Consistency (CON10):** 1 min accumulations present in the 10 min interval.

### 2.3.4 Continuous Rainfall Events

Rainfall events occurred consecutively or no more than 20 minutes after the previous one. Each continuous event has the following parameters:

- **Total duration of precipitation:** Interval between the start and end of precipitation in closed intervals of 10 min;

- **Effective duration of precipitation:** Only the time intervals (1 min resolution) in which there was actual precipitation within the total duration of the event;

- **Accumulated (ACE):** Volume of precipitation that occurred during the event;

- **Consistency (CONE):** Ratio between the number of time intervals (1 min) in which there was precipitation and the total duration of the event.

### 2.3.5 Cumulative RA drop

Drops in RA have been characterized using the following parameters:

– **Time after precipitation:** Time taken from the end of the event until the first drop in RA;

– **Duration of the RA drop:** Duration during which the RA drop persisted;

– **Average RA:** Average of RA's obtained during the drop.

## 2.4 Correlation between the wind at the lowest height of the profiler and the wind observed by the sonic anemometer at 10 m

Given that wind records during precipitation can have distorted values, Pearson's correlation has been used to analyze the representativeness of the wind speed data observed during the precipitation event. Pearson's coefficient was used to assess
whether there was any distortion between the wind speed observed by the anemometer of the micrometeorological tower located at a 10 m height and the wind speed observed by the equipment at its lowest operating height. If the correlation values were significantly lower during precipitation events compared to days without precipitation, this suggests that the precipitation interfered with the quality of the observations, impairing their representativeness. On the other hand, if the correlation values remained consistent regardless of the presence of precipitation, this indicates that the observations during precipitation events
are still representative and reliable.

Pearson's coefficient $r_{xy}$ is a single-value measure of the association between two variables, $x$ and $y$, being the ratio of their covariances to the product of their standard deviations, whose ideal value is equal to 1, as described in Equation 2:

$$r_{xy} = \frac{\frac{1}{n-1}\sum_{i=1}^{n}[(x_i - \bar{x})(y_i - \bar{y})]}{\sqrt{\frac{1}{n-1}\sum_{i=1}^{n}(x_i - \bar{x})^2}\sqrt{\frac{1}{n-1}\sum_{i=1}^{n}(y_i - \bar{y})^2}} \tag{2}$$

## 2.5 Analysis Steps

For SODAR, the analysis focused on the relationship between the consistency and accumulation of rainfall in 10-minute intervals and the corresponding range availability (RA). The average and standard deviation of the RA for each rainfall interval have been calculated, allowing for the identification of the rainfall conditions that exerted the most significant influence on the equipment's performance.

In addition, it was checked whether the observations made by SODAR during rainfall events remained representative or
were significantly affected by the rain. For this analysis, the wind speed recorded by the highest anemometer (at 10 m) of the micrometeorological mast was correlated with the wind speed estimated by the SODAR at its lowest range, at 30 m, comparing the data on days without precipitation with those obtained during precipitation events.

For LIDAR, the analyses were conducted separately for the periods of operation in regions near and far from the coast. The activities for the LIDAR near the coast followed the same steps as those applied to the SODAR. However, precipitation and
range availability (RA) were analyzed as continuous events rather than 10-minute intervals. This approach made it possible to assess the prolonged effects of precipitation on RA, considering the cumulative drops in RA after the events ended. Pearson's

correlation was applied between the various parameters monitored to verify the existence of dependent behaviors between them.

For such analysis, the wind speed recorded by the highest anemometer (at 10 m) of the micrometeorological mast was correlated with the wind speed measured by the LIDAR at its shortest range, at 40 m. For cases where the LIDAR is far from the coast, the same steps have been followed for the SODAR, with specific adaptations for the region's environmental conditions.

In addition to analyzing the influence of precipitation, secondary factors such as wind direction and cloudiness have also been analyzed since they may influence the profilers' performance. The wind direction's influence on RA was analyzed by grouping the directions into quadrants. Preference was given to the higher altitudes to determine the wind direction since range losses tend to start at these altitudes. In addition, the daily temperature variation was considered an indirect indicator for assessing cloudiness. The daily temperature variations were compared with a daily temperature model curve, allowing the correlation between temperature variations and the daily average RA to be analyzed.

## 3   Results

### 3.1   SODAR

During precipitation events, the SODAR's loss of range showed a correlation with CON10 and C10. However, the pattern observed was the fast recovery of range after the precipitation ended. In other words, measurements taken after the end of precipitation generally return to full functionality in the first sampling after the end of precipitation. Figure 2 depicts the equipment's range during operation when several precipitation events occurred, demonstrating the analyzed correlation. Rmax and RA tend to have similar values, but as explained in Section 2.3, while Rmax represents the longest range the equipment could estimate the wind speed, RA shows the loss ratio, which could happen at intermediate heights without affecting the maximum range. Therefore, to analyze the influence of rainfall on equipment availability, the point at which the CON10 value started to affect the RA was checked.

Figure 3a and Table 2 show the average RA for the events according to CON10. Figure 3a and Table 2 show that the SODAR had a variable RA even on days without precipitation, with an average of around 87%, due to other parameters that influence the range of the equipment, such as inhomogeneity of atmospheric temperature, air humidity, and ambient noise. It was also observed that the RA fell sharply for CON10 values from 7 min onwards, when average RA values were found to be below 50%, gradually reducing to close to zero.

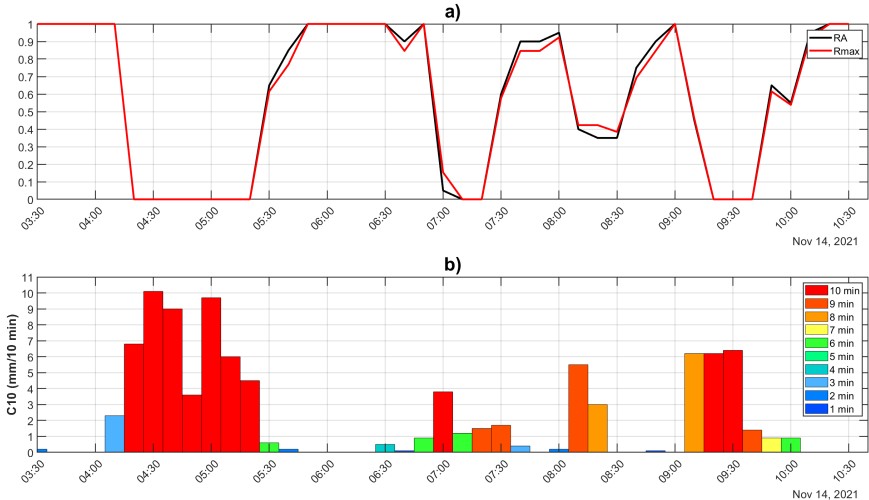

**Figure 2.** Figure 2a shows SODAR's Rmax (red line) and RA (black line) on November 14th, 2021, starting at 03:30 and ending at 10:30. Rmax and RA tend to have similar values, although slight differences can occur, as explained in item 2.3. It is important to note that Rmax is normalized, with the value of 1 corresponding to 260 m. In Figure 2b, the bars represent the C10 values, and the color of each bar represents the CON10 values.

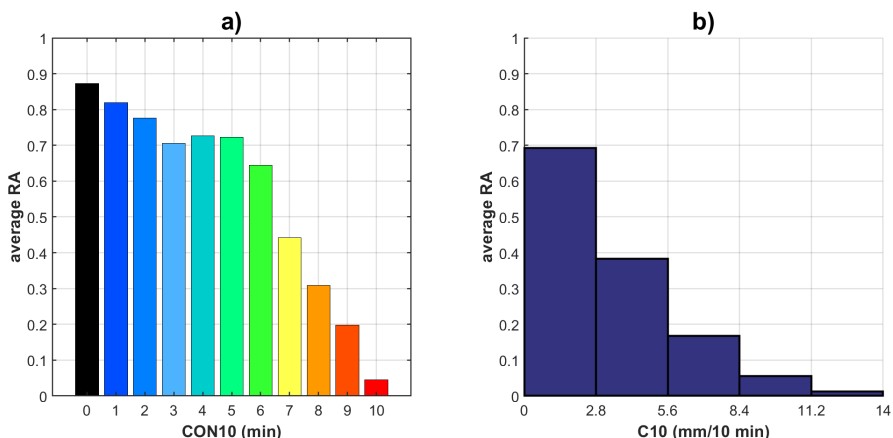

**Figure 3.** Figure 3a shows the average RA for each CON10 value ranging from 0 to 10 min. Figure 3b shows the average RA for C10 intervals between 0 and 14 mm/10 min, divided into five sub-intervals of the same size.

Considering such a result, only events with CON10 values of 7 min or more will be considered to analyze the SODAR's performance during precipitation events. Figure 3b and Table 3 show the average RA for the events according to the 5 C10 ranges. Figure 3b and data in Table 3 show that the SODAR dropped more than 50% in average RA for C10 values greater than 2.8 mm, gradually reducing as C10 increased.

**Table 2.** Average, Standard Deviation, and Coefficient of Variation of RA with CON10

| CON10 | 0 | 1 | 2 | 3 | 4 | 5 | 6 | 7 | 8 | 9 | 10 |
|---|---|---|---|---|---|---|---|---|---|---|---|
| Number of Events | 35,048 | 404 | 138 | 110 | 95 | 64 | 63 | 54 | 47 | 49 | 111 |
| Average RA | 0.873 | 0.819 | 0.776 | 0.706 | 0.726 | 0.723 | 0.644 | 0.442 | 0.309 | 0.198 | 0.045 |
| Standard Deviation | 0.211 | 0.260 | 0.289 | 0.329 | 0.349 | 0.341 | 0.370 | 0.410 | 0.342 | 0.298 | 0.154 |
| Coefficient of Variation | 24% | 32% | 37% | 47% | 48% | 47% | 57% | 93% | 111% | 150% | 345% |

**Table 3.** Average, Standard Deviation, and Coefficient of Variation of RA with C10.

| C10 | 0 to 2.8 | 2.9 to 5.6 | 5.7 to 8.4 | 8.5 to 11.2 | 11.3 to 14 |
|---|---|---|---|---|---|
| Number of events | 992 | 80 | 37 | 18 | 8 |
| Average RA | 0.692 | 0.383 | 0.167 | 0.055 | 0.012 |
| Standard Deviation | 0.359 | 0.410 | 0.348 | 0.235 | 0.035 |
| Coefficient of Variation | 52% | 107% | 208% | 424% | 283% |

Analyzing the values of the means and standard deviations of CON10 and C10, respectively, in Tables 2 and 3, the variation in range (RA) increases as the intensity and constancy of precipitation increase. Table 4 shows that in the case of constant precipitation (CON10 = 10 min), the loss of range is independent of the intensity of the precipitation. Situations where the CON10 is less than 10 allow measurements to be made without precipitation and integrated into the average values presented by SODAR. This feature is seen in the rise in the percentage values of the standard deviation, with the mean of each RA, as the values of CON10 and C10 increase.

**Table 4.** Average, Standard Deviation, and Coefficient of Variation of RA with C10 and CON10 = 10 min.

| C10 | 0 to 2.8 | 2.9 to 5.6 | 5.7 to 8.4 | 8.5 to 11.2 | 11.3 to 14 |
|---|---|---|---|---|---|
| Number of events | 27 | 32 | 30 | 16 | 6 |
| Average RA | 0.056 | 0.058 | 0.050 | 0 | 0.017 |
| Standard Deviation | 0.199 | 0.121 | 0.192 | 0 | 0.041 |
| Coefficient of Variation | 358% | 210% | 384% | 0% | 245% |

By separating the precipitation events with CON10 from 7 to 10 min, the RA values were also recorded for each event. Table 5 shows the times each RA value was recorded for each CON10. Table 5 shows that the highest concentration of RA recordings occurs near RA equal to 0 for the selected CON10 values (7, 8, 9, 10), starting at 33.33% for consistency 7 and gradually rising to 83.78% for consistency 10. There is a drop in values equal to or below 10% for the other RA values, except for RA equal to 1 and CON10 equal to 7. Based on this information, CON10 significantly impacts SODAR's performance. Precisely when the CON10 value reaches or exceeds 7 min.

**Table 5.** Distribution of SODAR RA at different CON10 values.

| CON10 | N° of Events | RA | | | | | | | | | | |
|---|---|---|---|---|---|---|---|---|---|---|---|---|
| | | 0 | 0.10 | 0.20 | 0.30 | 0.40 | 0.50 | 0.60 | 0.70 | 0.80 | 0.90 | 1.0 |
| 7 | 54 | **18** | 4 | 3 | 0 | 2 | 2 | 2 | 4 | 3 | 5 | 11 |
| | | **33.33%** | 7.41% | 5.56% | 0.00% | 3.70% | 3.70% | 3.70% | 7.41% | 5.56% | 9.26% | 20.37% |
| 8 | 47 | **19** | 2 | 4 | 4 | 1 | 2 | 5 | 3 | 1 | 2 | 4 |
| | | **40.43%** | 4.26% | 8.51% | 8.51% | 2.13% | 4.26% | 10.64% | 6.38% | 2.13% | 4.26% | 8.51% |
| 9 | 49 | **27** | 5 | 1 | 2 | 4 | 1 | 3 | 2 | 2 | 0 | 2 |
| | | **55.10%** | 10.20% | 2.04% | 4.08% | 8.16% | 2.04% | 6.12% | 4.08% | 4.08% | 0.00% | 4.08% |
| 10 | 111 | **93** | 8 | 2 | 4 | 1 | 1 | 0 | 0 | 0 | 0 | 2 |
| | | **83.78%** | 7.21% | 1.80% | 3.60% | 0.90% | 0.90% | 0.00% | 0.00% | 0.00% | 0.00% | 1.80% |

It was also checked whether the observations made by the SODAR during precipitation events were representative or whether they were affected by precipitation. To carry out this analysis, the correlation between the wind speed observed by the highest anemometer (10 m) of the micrometeorological mast and the wind speed observed by the SODAR at its lowest range (30 m) on days when there were no precipitation events and during precipitation events has been assessed.

Table 6 describes the number of events available to perform the correlation, and Figure 4 shows the distribution of these
events together with the Pearson correlation.

**Table 6.** Number of events observed for each situation.

| | |
|---|---|
| Events without rainfall | 12,773 |
| Events with rainfall | 741 |

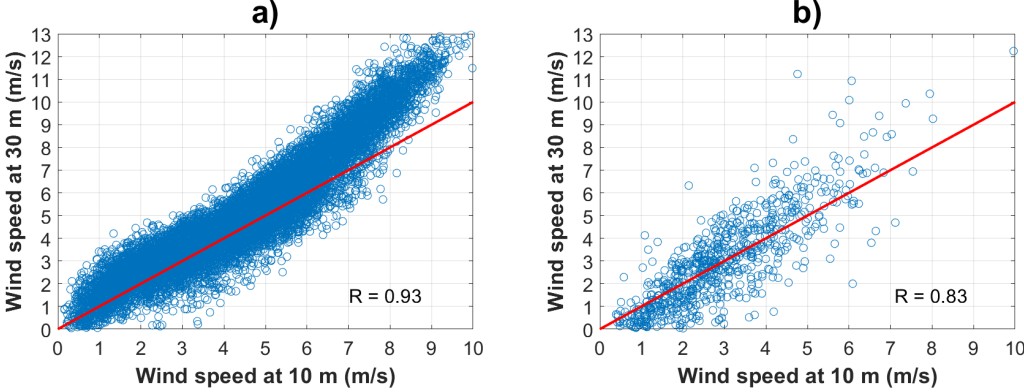

**Figure 4.** Figure 4a shows the dispersion of the wind speed at 10 m in relation to the wind speed at 30 m when there were no precipitation events. Figure 4b shows this same dispersion during precipitation events. The red line indicates the equality of the speeds of the anemometer and the wind profiler.

Considering that the anemometer is at the height of 10 m and the profiler's observation was made at 30 m, the profiler's values are expected to be faster than those of the anemometer.

The figures comparing wind speed at 10 m and 30 m revealed that without precipitation (Figure 4a), the correlation is very strong (R=0.93), indicating that the measurements are highly representative. During precipitation events (Figure 4b), the correlation decreases (R=0.83), showing that precipitation interferes with the measurements, increasing variability and reducing the reliability of the observations. The correlation is still strong, suggesting that measurements under precipitation conditions remain representative, albeit with more significant uncertainty.

The representativeness of SODAR measurements during precipitation events was assessed to determine whether the observations were consistent or affected by precipitation. Additional considerations were necessary to ensure the consistency of SODAR measurements during precipitation. An anemometer should be installed near the SODAR at a comparable height within its vertical profile. Since this setup was not feasible, the following approach was adopted: (1) the SODAR was installed in a flat area free of obstacles; (2) a correlation was assumed between the SODAR's lowest range (30 m) and a sonic anemometer at 10 m on a nearby micrometeorological tower; and (3) similar correlations under precipitation and non-precipitation conditions were used as evidence that precipitation did not degrade SODAR data quality. This analysis involved comparing the wind speed measured by the highest anemometer (10 m) on a micrometeorological mast with the wind speed observed by the SODAR at its lowest range (30 m). The analysis considered days with and without precipitation events. Table 6 presents the number of events analyzed, while Figure 4 illustrates the distribution of these events along with the Pearson correlation. As expected, due to the height difference, the SODAR values at 30 m were higher than those recorded by the anemometer at 10 m. Without precipitation (Figure 4a), the correlation was R=0.93, indicating highly representative measurements. During precipitation events (Figure 4b), the correlation decreased slightly to R=0.83, suggesting increased variability and reduced reliability. The Bias analysis revealed values of 0.65 for events without precipitation and 0.06 for events with precipitation, reflecting the expected difference in average wind speeds between 10 m and 30 m, mainly with high speed, that happens more frequently during good weather conditions. These findings support the conclusion that, while precipitation introduces some variability, SODAR measurements remain reliable and representative under such conditions.

The behavior of RA showed that SODAR is strongly affected by precipitation, but the return to typical RA occurs consistently soon after precipitation ends.

## 3.2   LIDAR

The LIDAR has operated in various locations, which makes it possible to analyze its performance in different environments. Thus, the two main situations analyzed were when the LIDAR was positioned near the coast and far from the coast. For the performance analysis near the coast, points P0, P1, P3, P6, and P7, located less than 8 km from the coast, have been selected. For the performance analysis further inland, points P4 and P5 were selected, which are located more than 20 km from the coast (see Table 1).

### 3.2.1 LIDAR near the coastline

The LIDAR's activities in this region during rainfall showed that the range has little correlation with CON10 in the same interval. It was noted that after a few sets of precipitation events (around 13%), there was a gradual range drop until the expected performance returned, as shown in Figure 5.

Notably, during the extreme rainfall event depicted in Figure 5, the LIDAR's performance was not affected when the event occurred, further demonstrating the robustness of the equipment under high-intensity precipitation conditions.

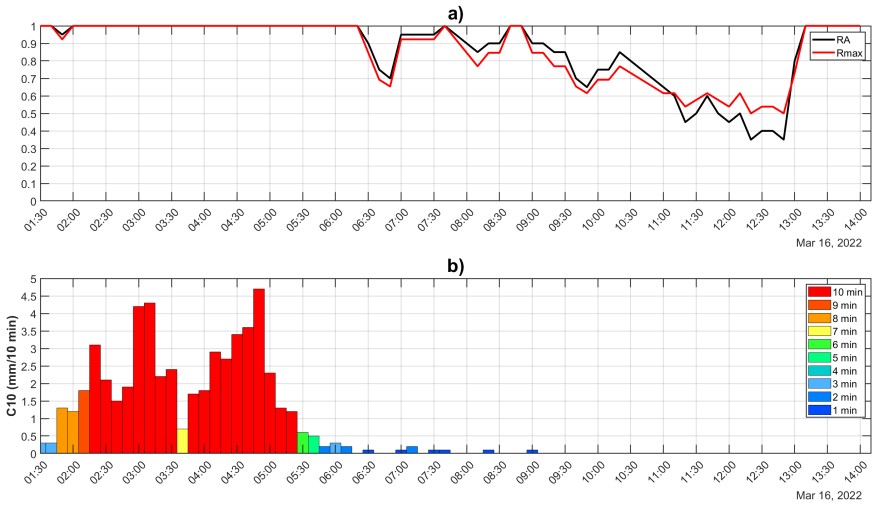

**Figure 5.** Figure 5a shows the equipment's Rmax (red line) and RA (black line) on March 16th, 2022, starting at 01:30 and ending at 14:00. Rmax and RA tend to have similar values, although slight differences can occur, as explained in item 2.3. It is important to note that Rmax is normalized, with the value of 1 corresponding to 260 m. In Figure 5b, the bars represent the C10 values, and the color of each bar represents the CON10 values.

To reinforce the assertion that range has little correlation with rainfall at the time of its occurrence, the influence of rainfall on the DA of the equipment was analyzed according to its CON10 and C10. Figure 6, Table 7, and Table 8 show the average RA for the events according to their CON10 and C10. These graphs and tables show minimal statistical variation, indicating that rainfall has no influence on RA during rainfall events.

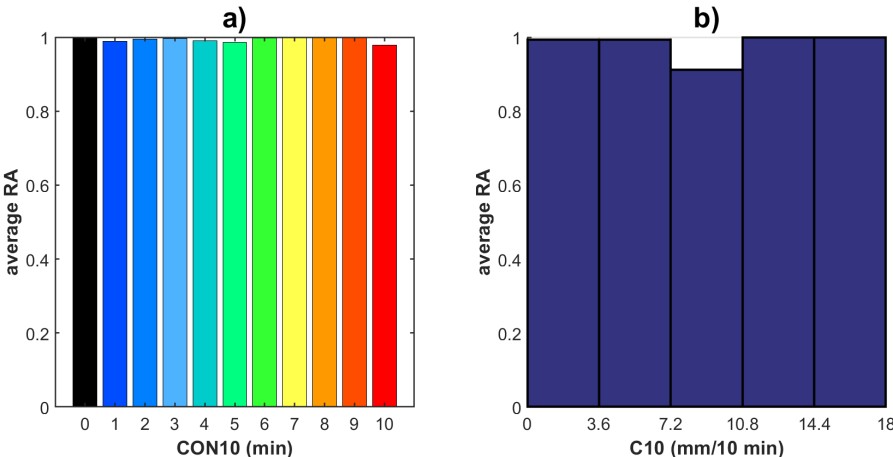

**Figure 6.** Figure 6a shows the average RA for each CON10 value from 0 to 10 min. Figure 6b shows the average RA for C10 intervals between 0 and 18 mm/10 min, divided into five subintervals of the same size.

**Table 7.** Average, Standard Deviation, and Coefficient of Variation of RA with CON10.

| CON10 | 0 | 1 | 2 | 3 | 4 | 5 | 6 | 7 | 8 | 9 | 10 |
|---|---|---|---|---|---|---|---|---|---|---|---|
| Number of Events | 38671 | 287 | 118 | 81 | 58 | 43 | 43 | 42 | 39 | 40 | 78 |
| Average RA | 0.998 | 0.989 | 0.995 | 0.996 | 0.991 | 0.986 | 0.998 | 1.000 | 0.999 | 0.999 | 0.978 |
| Standard Deviation | 0.026 | 0.045 | 0.030 | 0.025 | 0.072 | 0.064 | 0.011 | 0.000 | 0.008 | 0.008 | 0.116 |
| Coefficient of Variation | 3% | 5% | 3% | 2% | 7% | 6% | 1% | 0% | 1% | 1% | 12% |

**Table 8.** Average, Standard Deviation, and Coefficient of Variation of RA with C10.

| C10 | 0 to 3.6 | 3.6 to 7.2 | 7.2 to 10.8 | 10.8 to 14.4 | 14.4 to 18 |
|---|---|---|---|---|---|
| Number of Events | 768 | 37 | 16 | 6 | 2 |
| Average RA | 0.993 | 0.993 | 0.913 | 1.000 | 1.000 |
| Standard Deviation | 0.040 | 0.041 | 0.242 | 0.000 | 0.000 |
| Coefficient of Variation | 4% | 4% | 27% | 0% | 0% |

To check whether the observations made by the LIDAR during precipitation events were representative, the Pearson correlation of the wind speed observed by the highest anemometer (10 m) at the micrometeorological station with the wind speed observed by the LIDAR at its lowest range (40 m) on days when there were no precipitation events and during precipitation events have been assessed.

The number of observations available to carry out the correlation is described in Table 9. The distribution for these events, together with the Pearson correlation (0.96), showed that there was no decrease in the correlation between events with and without precipitation, as shown in Figure 7.

The representativeness of LIDAR measurements during precipitation events was assessed to determine whether the obser-
vations were consistent or affected by precipitation. Additional considerations were necessary to ensure the consistency of
LIDAR measurements during precipitation. An anemometer should be installed near the LIDAR at a comparable height within
its vertical profile. Since this setup was not feasible, the following approach was adopted: (1) the LIDAR was installed in a flat
area free of obstacles; (2) a correlation was assumed between the LIDAR's lowest range (40 m) and a sonic anemometer at
10 m on a nearby micrometeorological tower; and (3) similar correlations under precipitation and non-precipitation conditions
were used as evidence that precipitation did not degrade LIDAR data quality.

     This analysis involved comparing the wind speed measured by the highest anemometer (10 m) on a micrometeorological
mast with the wind speed observed by the LIDAR at its lowest range (40 m). The analysis considered days with and without
precipitation events. Table 9 presents the number of events analyzed, while Figure 7 shows the distribution of these events
along with the Pearson correlation.

     As expected, the LIDAR values at 40 m were higher than those recorded by the anemometer at 10 m due to the height
difference. The distribution for these events, together with the Pearson correlation of 0.96, shows no decrease in the correlation
between events with and without precipitation, as shown in Figure 7. The Bias analysis revealed values of 1.80 for events
without precipitation and 1.48 for events with precipitation, reflecting the expected difference in average wind speeds between
10 m and 40 m. These findings support the conclusion that LIDAR measurements remain reliable and representative under
such conditions.

**Table 9.** Number of events observed for each situation.

| Events without rainfall | 12,094 |
| --- | --- |
| Events with rainfall | 498 |

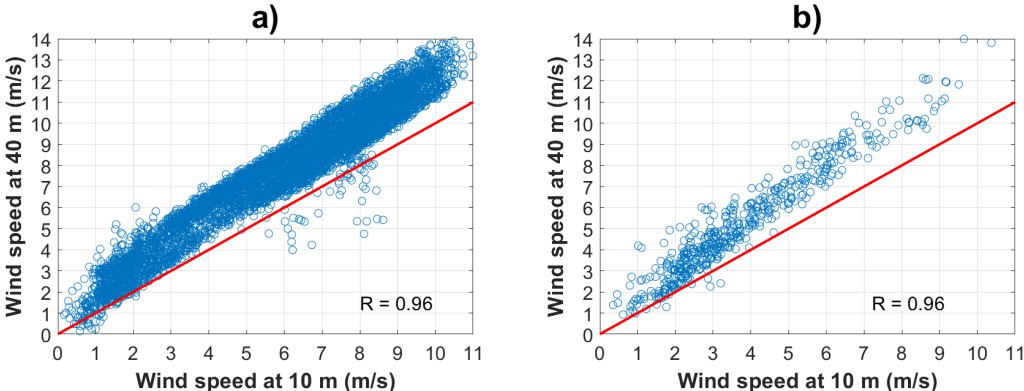

**Figure 7.** Figure 7a shows the dispersion of the wind speed at 10 m in relation to the wind speed at 40 m when there are no precipitation
events. Figure 7b shows this same dispersion during precipitation events. The red line indicates the equality of the speeds of the anemometer
and the profiler.

Continuous rainfall events and accumulated RA drops were considered to analyze the effects of events on RA after they have ended. The events in which there was no drop after precipitation (87%) were separated from those in which there was a drop in RA (13%), as shown in Table 10.

Next, the events were classified according to consistency, duration, and accumulation, indicating how many did not result in a drop in RA and how many resulted in a drop after they occurred, as shown in Tables 11, 12, and 13. It was found that, for most of the events in each subdivision, the proportion of those that showed a drop in RA after they occurred was below 40% of the total events, as illustrated in Figure 8.

**Table 10.** Number of events in which RA dropped after precipitation.

| No RA drop | RA dropped after rainfall | Total |
|---|---|---|
| 187 (87%) | 27(13%) | 214 (100%) |

**Table 11.** Number of events with a drop in RA in relation to CONE.

| CONE (%) | No RA drop | RA dropped after rainfall | Total |
|---|---|---|---|
| 10 | 89 (96%) | 4 (4%) | 93 |
| 20 | 23 (92%) | 2 (8%) | 25 |
| 30 | 26 (84%) | 5 (16%) | 31 |
| 40 | 15 (75%) | 5 (25%) | 20 |
| 50 | 14 (82%) | 3 (18%) | 17 |
| 60 | 8 (80%) | 2 (20%) | 10 |
| 70 | 7 (64%) | 4 (36%) | 11 |
| 80 | 5 (83%) | 1 (17%) | 6 |
| 90 | 0 (0%) | 1 (100%) | 1 |
| 100 | 0 | 0 | 0 |

**Table 12.** Number of events with RA drop in relation to the total duration.

| Rainfall duration (min) | No RA drop | RA dropped after rainfall | Total |
|---|---|---|---|
| 10 | 97 (95%) | 5 (5%) | 102 |
| 20 | 35 (81%) | 8 (19%) | 43 |
| 30 | 16 (94%) | 1 (6%) | 17 |
| 40 | 8 (89%) | 1 (11%) | 9 |
| 50 | 9 (75%) | 3 (25%) | 12 |
| 60 | 4 (67%) | 2 (33%) | 6 |
| 70 | 7 (100%) | 0 (0%) | 7 |
| 80 | 3 (75%) | 1 (25%) | 4 |
| 90 | 2 (67%) | 1 (33%) | 3 |
| $\geq$100 | 6 (55%) | 5 (45%) | 11 |

**Table 13.** Number of events with a drop in RA in relation to the average ACE.

| Average ACE (mm/min) | No RA drop | RA dropped after rainfall | Total |
|---|---|---|---|
| $0 < x \leq 0.05$ | 128 (93%) | 10 (7%) | 138 |
| $0.05 < x \leq 0.10$ | 23 (79%) | 6 (21%) | 29 |
| $0.10 < x \leq 0.15$ | 14 (82%) | 3 (18%) | 17 |
| $0.15 < x \leq 0.20$ | 7 (58%) | 5 (42%) | 12 |
| $0.20 < x \leq 0.25$ | 6 (100%) | 0 (0%) | 6 |
| $0.25 < x \leq 0.30$ | 4 (67%) | 2 (33%) | 6 |
| $0.30 < x \leq 0.35$ | 1 (100%) | 0 (0%) | 1 |
| $0.35 < x \leq 0.40$ | 0 (0%) | 0 (0%) | 0 |
| $0.40 < x$ | 4 (80%) | 1 (20%) | 5 |

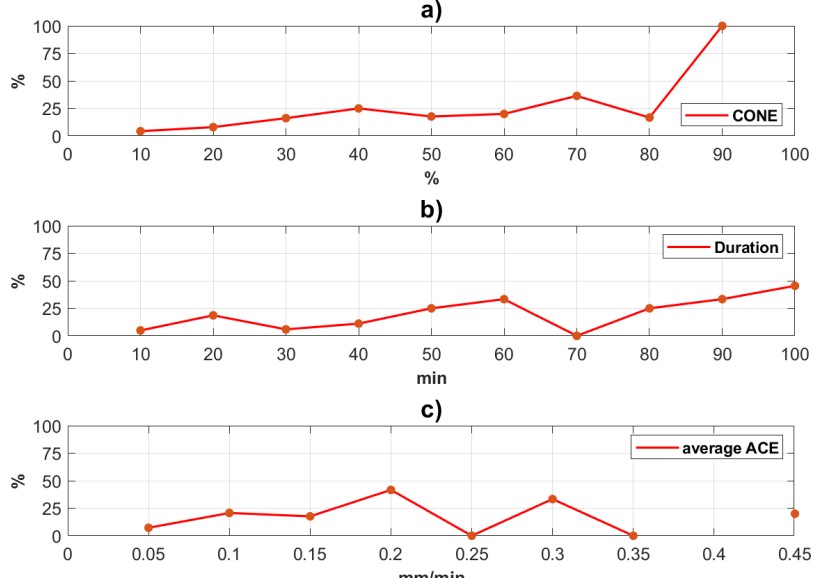

**Figure 8.** Figure 8a shows the occurrences of falling RA after precipitation as a function of CONE. Figure 8b shows the occurrences of falling RA after precipitation as a function of the duration of precipitation. Figure 8c shows the occurrences of falling RA after precipitation as a function of average ACE.

Pearson's correlation was applied to the various parameters to check for dependent behavior between them. The precipitation parameters showed a low correlation with the RA drop parameters, except the duration of the event with the duration of the RA drop, which showed a correlation of 0.734, as shown in Table 14.

**Table 14.** Correlation between precipitation event parameters and RA drop parameters.

|  | The total duration of rainfall | Effective rainfall duration | ACE | CONE |
|---|---|---|---|---|
| Time after rainfall | -0.068 | -0.150 | -0.182 | -0.211 |
| Duration of RA drop | **0.734** | **0.847** | 0.641 | 0.309 |
| Average RA | -0.078 | -0.179 | -0.172 | -0.526 |

Based on the results obtained in the analysis of the LIDAR's performance during precipitation events, it can be concluded that its performance was not compromised during these weather conditions. Both the range and accuracy of the equipment remained consistent during precipitation. Furthermore, the drops observed after precipitation events were not significant enough (Table 10) to suggest an influence of precipitation. One possible explanation for the equipment's high performance is its location close to the coast, where the presence of marine aerosols is abundant, providing consistent targets for measurements most of the time. The aerosols quickly recovered after a rain event, not significantly affecting the LIDAR's operation.

### 3.2.2 LIDAR far from the coastline

While the LIDAR far from the coast has been observed to behave quite differently from the LIDAR near the coast, variations in RA before, during, and after precipitation have been observed even in periods without precipitation events nearby, due to conditions that will be detailed below.

Based on the analysis of Figure 9, Table 15, and Table 16, which show the average RA for the events as a function of their CON10 and C10, rainfall influences the average RA. Although the average RA with rainfall is lower than the average without rainfall, it was not possible to establish a correlation between this drop and the different levels of CON10 or C10.

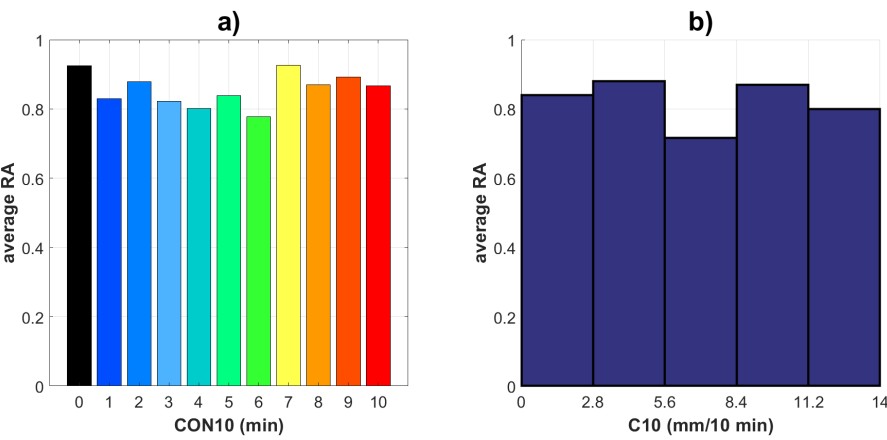

**Figure 9.** Figure 9a shows the average RA for each CON10 value ranging from 0 to 10 min. Figure 9b shows the average RA for C10 intervals between 0 and 14 mm/ 10 min, divided into five sub-intervals of the same size.

**Table 15.** Average, Standard Deviation, and Coefficient of Variation of RA with CON10.

| CON10 | 0 | 1 | 2 | 3 | 4 | 5 | 6 | 7 | 8 | 9 | 10 |
|---|---|---|---|---|---|---|---|---|---|---|---|
| Number of Events | 11,385 | 114 | 54 | 48 | 32 | 29 | 22 | 15 | 13 | 14 | 27 |
| Average RA | 0.924 | 0.829 | 0.879 | 0.822 | 0.802 | 0.838 | 0.777 | 0.927 | 0.869 | 0.893 | 0.867 |
| Standard Deviation | 0.207 | 0.257 | 0.205 | 0.300 | 0.299 | 0.270 | 0.312 | 0.143 | 0.247 | 0.223 | 0.159 |
| Coefficient of Variation | 22% | 31% | 23% | 36% | 37% | 32% | 40% | 15% | 28% | 25% | 18% |

**Table 16.** Average, Standard Deviation, and Coefficient of Variation of RA with C10.

| C10 | 0 to 2.8 | 2.8 to 5.6 | 5.6 to 8.4 | 8.4 to 11.2 | 11.2 to 14 |
|---|---|---|---|---|---|
| Number of Events | 330 | 25 | 6 | 5 | 2 |
| Average RA | 0.840 | 0.880 | 0.717 | 0.870 | 0.800 |
| Standard Deviation | 0.254 | 0.231 | 0.317 | 0.291 | 0.283 |
| Coefficient of Variation | 30% | 26% | 44% | 33% | 35% |

The Pearson correlation has been used to compare the wind speed observed by the highest anemometer (10 m) at the micrometeorological mast with the wind speed observed by the LIDAR at its lowest range (40 m) on days when there were no precipitation events and during precipitation events. The number of observations available for the correlation is shown in Table 17. Analysis of the distribution of these events, together with Pearson's correlation, revealed no significant reduction in the correlation between events with and without precipitation, as illustrated in Figure 10.

The representativeness of LIDAR measurements during precipitation events was assessed to determine whether the observations were consistent or affected by precipitation. Additional considerations were necessary. An anemometer should be installed near the LIDAR at a comparable height within its vertical profile. Since this setup was not feasible, the following approach was adopted: (1) the LIDAR was installed in a flat area free of obstacles; (2) a correlation was assumed between the LIDAR's lowest range (40 m) and a sonic anemometer at 10 m on a nearby micrometeorological tower; and (3) similar corre-

lations under precipitation and non-precipitation conditions were used as evidence that precipitation did not degrade LIDAR data quality.

This analysis involved comparing the wind speed measured by the highest anemometer (10 m) on a micrometeorological mast with the wind speed observed by the LIDAR at its lowest range (40 m). The analysis considered days with and without precipitation events. Table 17 presents the number of events analyzed, while Figure 10 illustrates the distribution of these events

along with the Pearson correlation.

Analysis of the distribution of these events, with Pearson's correlation, revealed no significant change in the correlation between events with (0.93) and without precipitation (0.90), as illustrated in Figure 10. The analysis revealed Bias values of 1.65 for events without precipitation and 1.31 for events with precipitation, reflecting the expected difference in average

**Table 17.** Number of events observed for each situation.

| | |
|---|---|
| Events without rainfall | 3,378 |
| Events with rainfall | 221 |

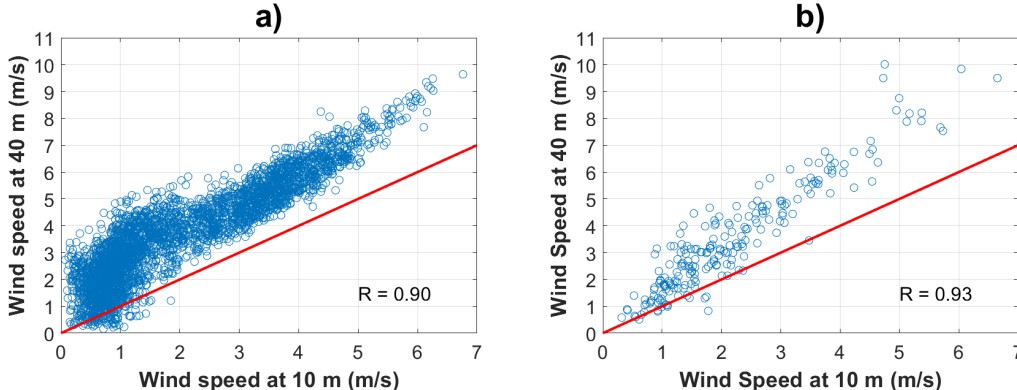

**Figure 10.** Figure 10a shows the dispersion of the wind speed at 10 m in relation to the wind speed at 30 m when there were no precipitation events. Figure 10b shows the same dispersion but during precipitation events. The red line indicates the equality of the speeds of the anemometer and the wind profiler. Source: Author

wind speeds between 10 m and 40 m. These findings support the conclusion that LIDAR measurements remain reliable and representative under such conditions.

Considering that the sonic anemometer is at 10 m and the wind profiler observation was made at 40 m, an overspeed of the wind profiler values compared to those of the anemometer is to be expected.

The analysis of RA drops after precipitation was not carried out because the experiments occurred during the dry season.

In general, the LIDAR RA far from the coast compared to near the coast was more variable, both on rainy and clear days, as shown in Table 18, which contains the average RA for the days when there was precipitation, for the days when there was no precipitation and for the days when there was precipitation. However, there was a drop in RA.

**Table 18.** Average RA on the days when there was precipitation, no precipitation, and for the days when there was no precipitation and a drop in RA.

| | | With rainfall | Without rainfall | RA drop without rainfall | Total |
|---|---|---|---|---|---|
| | Number of Days | 86 (32.21%) | 181 (67.79%) | 15 (5.62%) | 267 (100.00%) |
| Near the coastline | Average | 0.997 | 0.999 | 0.982 | 0.998 |
| | Standard Deviation | 0.031 | 0.024 | 0.081 | 0.026 |
| | Number of days | 37 (49.33%) | 38 (50.67%) | 18 (24.00%) | 75 (100.00%) |
| Far from the coastline | Average | 0.856 | 0.981 | 0.961 | 0.918 |
| | Standard Deviation | 0.273 | 0.092 | 0.128 | 0.214 |

Figure 11 shows the behavior of the LIDAR RA during a period in which several precipitation events occurred. The RA was already falling before the precipitation events, which made us question whether the precipitations were primarily responsible for these falls.

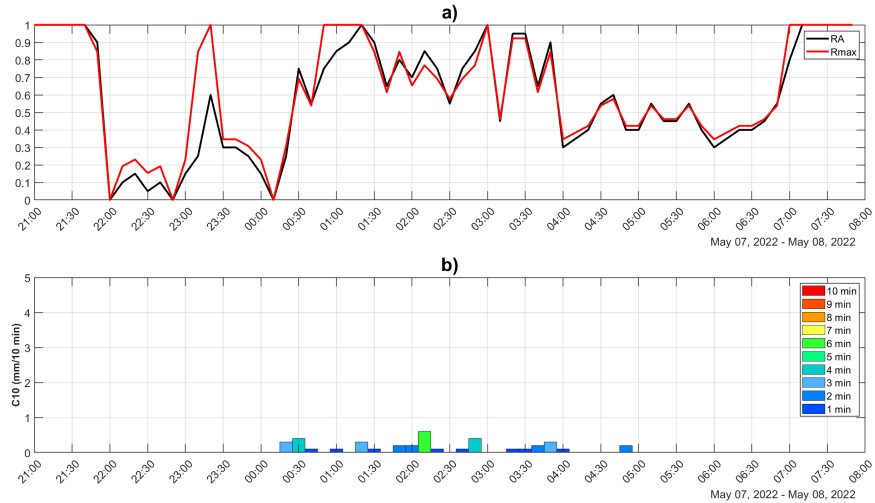

**Figure 11.** Figure 11a shows the equipment's Rmax (red line) and RA (black line) on May 7th, 2022, starting at 01:30 and ending at 14:30. Rmax and RA tend to have similar values, although slight differences can occur, as explained in item 2.3. It is important to note that Rmax is normalized, with the value of 1 corresponding to 260 m. In Figure 11b, the length of the bars represents the C10 values and the color of each bar represents the CON10 values.

In an attempt to find another variable that could influence the behavior of the LIDAR's RA, wind direction and atmospheric cloudiness were analyzed as factors that could influence the equipment's RA in addition to precipitation.

Regarding wind direction, it was hypothesized that winds without a marine influence could transport fewer aerosols or that the aerosols transported could be less efficient LIDAR targets, causing a reduction in RA.

Cloudiness, which influences air temperature variation, was hypothesized to interfere with turbulent flow, reducing the amount of aerosols in suspension and consequently affecting the RA.

## 3.3 Analysis of Horizontal Wind Direction

The wind directions were grouped into quadrants with the following intervals to analyze the influence of wind direction on the RA of the equipment:

– 1st quadrant: equal to 0° to less than 90°

– 2nd quadrant: equal to 90° to less than 180°

– 3rd quadrant: equal to 180° to less than 270°

– 4th quadrant: equal to 270° to less than 360°

The highest height at which the wind was observed has been used to determine the wind direction, as it is generally at these heights that the loss of range begins. Table 18 shows the average RA for each wind direction quadrant for both the LIDAR near the coast and the LIDAR far from the coast.

Table 19 shows that when the LIDAR was close to the coast, wind direction observations were predominantly in the first quadrant, with 89.63% of the wind profiles observed, representing the largest share of winds coming from the sea. The average RA is high in all quadrants, ranging from 0.94 to 0.99, suggesting good range availability regardless of wind direction.

For the LIDAR far from the coast, winds still predominate in the first quadrant (69.74%), but the distribution expands slightly to the second quadrant (25.86%). The number of winds in the third and fourth quadrants increased compared to the LIDAR near the coast, indicating more significant variability in wind direction when further from the coast.

The average RA was high in the first and second quadrants (0.95) and (0.89), respectively, but it sharply dropped in the third and fourth quadrants (0.52) and (0.61), suggesting that inland winds have a significant impact on the equipment's range. To support such a conclusion, the authors performed at a point close to the coast (P3). The point where the most intense rainy period occurred in the entire experiment (9.78 mm/day) was chosen. The rainy period results in a decrease in the influence of the trade winds and the intensity of the sea breeze. These two conditions lead to weaker winds and longer periods in directions coming from the continent, resulting in less influence at P3 from maritime aerosols, similar to points far from the coast. Eighteen rain events were found when analyzing this point, ten of which had RA drops. Of these, seven had speeds below 5 m/s, three had speeds between 7 and 5 m/s, seven had direction coming from the sea, and three had direction coming from the land. There were also eight precipitation events without RA drops. In all of them, the wind direction came from the sea, and the speed was above 5 m/s. This analysis reinforces the understanding that low-intensity wind or wind from the continent delays the return of aerosols responsible for recovering RA performance.

**Table 19.** Average RA for each quadrant of wind direction.

| Average RA for each wind direction quadrant | | | | | | |
|---|---|---|---|---|---|---|
| Position relative to the coastline | Quadrants | 1° | 2° | 3° | 4° | Total |
| Near | Number of events | 34,237 (89.63%) | 3,483 (9.12%) | 227 (0.59%) | 253 (0.66%) | 38,200 (100.00%) |
| | Average RA | 0.999 | 0.994 | 0.942 | 0.986 | - |
| | Standard Deviation | 0.020 | 0.043 | 0.138 | 0.075 | - |
| Far | Number of events | 7,395 (69.74%) | 2,742 (25.86%) | 318 (3.00%) | 148 (1.40%) | 10,603 (100.00%) |
| | Average RA | 0.950 | 0.892 | **0.521** | **0.610** | - |
| | Standard Deviation | 0.172 | 0.223 | 0.362 | 0.327 | - |

## 3.4 Cloudiness Analysis

For the cloudiness analysis, the adopted approach considered the daily temperature variations as indicators of cloudiness and thus compared them with the daily average RA. To obtain this daily temperature variation, a daily temperature curve model

was used to compare with other observed temperature curves. The daily temperature curve model chosen was for July 25th, 2022, as shown in Figure 12. This curve was smoothed using the moving average statistical technique, in which the average of the neighboring points replaces each point in the data series.

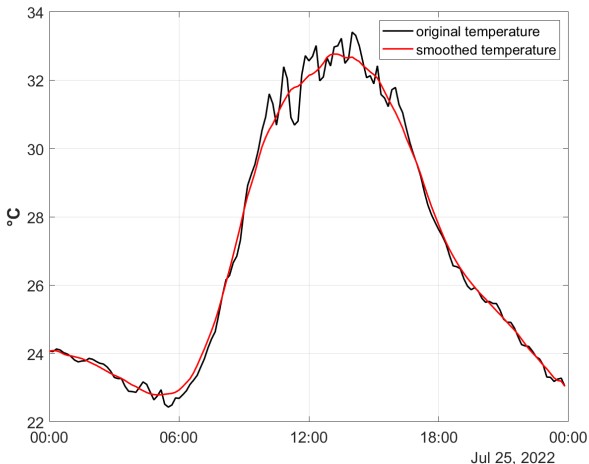

**Figure 12.** Figure 12 shows the July 25th, 2022, temperature curve (black line) and the smoothed temperature curve (red line) for the same day.

Using this curve model, the average modulus of the differences between the temperature curve model and the daily observed temperature curves has been assessed, thus forming the cloudiness indicator. Pearson's correlation between this cloudiness indicator and the daily average RA was used for comparison. The correlation is equal -0.3531, indicating that although the coefficient is negative, it has a low value. This suggests a weak influence between these two variables, meaning that the cloudiness indicator has no influence on the equipment's daily RA.

## 4   Conclusions

It was observed that the SODAR, operating at a single point, showed a fast recovery of range availability (RA) after the end of precipitation. CON10 and C10 significantly impacted the SODAR's RA (average RA less than 50% from 7 min of consistency). However, measurements in rainy conditions continued to be reasonably representative (high wind speed correlations both for days without rain (0.93) and for periods with rain (0.83)). On the other hand, the LIDAR, operating both near and far from the coast, showed variations in behavior. Near the coast, rainfall (CON10 and C10) did not instantly influence RA (average DA of 0.97 for 10 min CON10), and the measurements remained representative (high wind speed correlations for both days without rain (0.96) and periods with rain (0.96)). The drops in RA after precipitation occur only in 13% of all events, suggesting there is no substantial influence.

Further away from the coast, the LIDAR showed variations in RA before, during, and after precipitation, even in periods with no nearby rainfall. In these conditions, RA did not decrease proportionally with increasing CON10 or C10, and measurements in precipitation conditions remained representative (high wind speed correlations for both days without rain (0.90) and periods with rain (0.93)). Inland winds significantly impacted the equipment's range (average RA close to 50%), while the cloudiness indicator did not significantly affect daily RA, given the correlation coefficient $R = -0.35$.

While the SODAR worked at a single point, the LIDAR operated at several locations, revealing different behaviors depending on their proximity to the coast. Therefore, location and specific weather conditions must be considered when using these technologies for atmospheric measurements. Given the difference in the LIDAR performance when there is no sea breeze, it is recommended that observations be made over extended periods in urban and rural areas where the marine component is not present in order to compare the statistics found in this work, which analyzed a limited number of cases. On the other hand, the SODAR's performance may have been reduced due to its proximity to the coast, considering that the sea breeze softens the temperature variation, the main phenomenon in generating sound pulse backscatter. Therefore, analyzing the SODAR's performance in a continental environment with more thermal variation could consolidate this hypothesis.

*Data availability.* The data used in this investigation are available in https://eosolar.equatorialenergia.com.br/. Equatorial Energia and Gera Maranhão must be cited.

*Author contributions.* Adriel Carvalho performed Data Curation, Formal Analysis, Investigation, and Visualization. Francisco Albuquerque contributed with Conceptualization, Investigation, Methodology, Supervision, Validation, Writing - review. Denisson Oliveira contributed with Funding Acquisition, Project Administration, Resources, Writing - original draft and review.

*Competing interests.* The authors declare that they have no conflict of interest.

*Acknowledgements.* The authors thank to Coordenação de Pessoal de Ensino Superior (CAPES), Brazil, for partially funding this investigation under grants CAPES 88881.707433/2022-01 and 88887.707432/2022-00. The authors thank to Fundação de Âmparo à Pesquisa e ao Desenvolvimento Científico e Tecnológico do Maranhão (FAPEMA) for funding this investigation/publication.

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
