# Peer review of "ANALYZING THE PERFORMANCE OF VERTICAL WIND PROFILERS IN RAIN EVENTS"

_Wind Energy Science, 2024_

## Referee Comment (RC1)

Review of the paper "ANALYZING THE PERFORMANCE OF VERTICAL WIND PROFILERS IN RAIN EVENTS ANALYZING THE PERFORMANCE OF VERTICAL WIND PROFILERS IN RAIN EVENTS" by Carvalho et. Al.

**General comment**

The present paper deals with an evaluation of the availability of SODAR and LIDAR measurements in case of rain. Since LIDAN and SODAR devices are nowadays essential for many activities related to the wind energy discipline (e.g. site assessment, field testing, etc..), the topic of this work deserves investigation.

The adopted methodology is sound, and the analyzed dataset is fit for purpose.

The paper is well written, even if I found many points that should be clarified before the acceptance of the paper. All these points are listed in the "Comments" section, here below.

**Comments**

1. Table 1: please, consider the possibility of appending a geographic representation of the measurement points to the table itself.
2. Section 2.3: Instead of using subsubsections, it could be useful to describe all criteria using a bulleted list.
3. Figure 1: when commenting on Figure 1, is it possible to spend a few words to explain the fact that $R_{max}$ may be greater or lower than RA? This was just mentioned in section 2.3, but here a clarification could help readers understand the employed criteria.
4. Figure 2 (and many others): In the caption, it is written "Source: Author". What does it mean?
5. Figure 3: This figure is hard to interpret:
    a. The legend can be improved. I guess the different lines refer to different levels of C10.
    b. Y-label can be ameliorated.
6. General comments on section 3.1:
    a. The values of Standard deviations in Tables 2 and 3 deserve explanations. They are mildly commented on but may bear important information.
    b. There is some kind of mixed relationship between CON10 and C10... What happens for high C10 but low CON10 and vice versa? Maybe a double-entry table or a 3D plot may help readers understand the characteristics of RA combining these two criteria.
7. Figure 4 and related comments: It is hard to extract some pieces of information from this analysis because the wind at different altitudes is typically different due to the atmospheric shear layer. Additionally, different shear layers can be experienced in the same location according to the stability of the atmosphere. I strongly suggest either

eliminating the plots (and the few lines of comments) from the text, or extending thoroughly the analysis as, in the current state, the plots does not offer a valuable output given the scope of the paper.

8. Figure 7: same observation as the previous point.

9. Figure 8: missing x-label. Y-label can be improved.

10. Pag.17, lines 254-255: As it was previously written, Here the overspeed is due to the shear layer. Since we might experience shear layers of different magnitudes, the difference between measurements taken at 10m and 40m cannot be compared directly. Hence, fig 10 and the related comments hardly provide useful information. Please, clarify.

11. Line 258: "suggesting that inland winds have a significant impact on the equipment's range", this needs a justification, otherwise it could be viewed as a spurious correlation. Could it be dependent on wind events, that may occur more often in the case of inland winds?

---

## Author Comment (AC1)

Ref.: Article WES-2024-132 "Analyzing the performance of vertical wind profilers in rain events."
Author(s): Adriel J. Carvalho et al.
MS type: Research article."

Dear Editor,

We have attached the answers to the reviewer's comments regarding Adriel J. Carvalho et al.'s paper "Analyzing the performance of vertical wind profilers in rain events." The answers are in red, and the modifications are indicated in red in the manuscript.

Best wishes,

Denisson Q. Oliveira
Federal University of Maranhão, Brazil

ANSWERS TO REVIEWER 1

The authors appreciate the reviewers' valuable comments. We hope that the answers match all the concerns raised about the manuscript. All modifications are indicated below.

1. Table 1: Please, consider the possibility of appending a geographic representation of the measurement points to the table itself.

The authors have included a new figure (Figure 1) in the manuscript. The other figures have been re-numbered according to the modifications.

2. Section 2.3: Instead of using subsubsections, it could be useful to describe all criteria using a bulleted list.

The authors analyzed the suggestion but did not follow it because section 2.3 already has many bullets in subsections. Changing the structure using a bulleted list may not help to improve the manuscript.

3. Figure 1: when commenting on Figure 1, is it possible to spend a few words to explain the fact that Rmax may be greater or lower than RA? This was just mentioned in section 2.3, but here a clarification could help readers understand the employed criteria.

As requested, the authors have included additional comments regarding Rmax and RA in Section 3.1. The following text has been included: "Rmax and RA tend to have similar values, but as explained in Section 2.3, while Rmax represents the longest range the equipment could estimate the wind speed, RA shows the loss ratio, which could happen at intermediate heights, without affecting the maximum range." Figure 2 caption has also been improved.

4. Figure 2 (and many others): In the caption, it is written "Source: Author". What does it mean?

The phrase "Source: Author" has been inserted into captions, meaning the figure is created by the authors (not from other references). The term "Source: Author" has been removed from all figures to avoid misunderstanding.

5.Figure 3: This figure is hard to interpret:
   a. The legend can be improved. I guess the different lines refer to different levels of C10.
   b. Y-label can be ameliorated.

The authors agree with the reviewer that the figure did not help the reader understand the subject. To improve the manuscript and to describe the results from Figure 3, it has been changed by Table 4, which brings the same information differently. A new explanation was provided in the text referencing Table 4, which deals with the same subject.

6. General comments on section 3.1:
   a. The values of Standard deviations in Tables 2 and 3 deserve explanations. They are mildly commented on but may bear important information.

Section 3.1 includes additional text discussing the behavior of the standard deviation of CON 10 and C10, with their averages. Tables 2 and 3 have been updated according to the reviewer's suggestion.
Below is the text included in the article:
"Analyzing the values of the means and standard deviations of CON10 and C10, respectively, in Tables 2 and 3, it can be seen that the variation in range (RA) increases as the intensity and constancy of precipitation increase. This trend is seen in the rise in the percentage values of the standard deviation, with the mean of each RA, as the values of CON10 and C10 increase."

   b.
There is some kind of mixed relationship between CON10 and C10... What happens for high C10 but low CON10 and vice versa? Maybe a double-entry table or a 3D plot may help readers understand the characteristics of RA combining these two criteria.

To address the reviewer's question, the different rainfall intensities (C10) were compared with the consistency of rainfall (CON10). The drop in RA was found to be more related to CON10 than C10 since, with high CON10; the RA reduction occurs for all C10 levels. As CON10 decreases, there is a smaller reduction in RA, although higher C10 values have some influence on RA for CON10 values lower than 10. A table showing RA values (mean, standard deviation, and coefficient of variation) with various C10 intensities and the maximum CON10 value was added and discussed.

Table 4 – Average, Standard Deviation, and Coefficient of Variation of RA with C10 and CON10 = 10 min

| C10 | 0 to 2.8 | 2.9 to 5.6 | 5.7 to 8.4 | 8.5 to 11.2 | 11.3 to 14 |
|---|---|---|---|---|---|
| Number of events | 27 | 32 | 30 | 16 | 6 |
| Average RA | 0.056 | 0.058 | 0.050 | 0 | 0.017 |
| Standard Deviation | 0.199 | 0.121 | 0.192 | 0 | 0.041 |
| Coefficient of Variation | 358% | 210% | 384% | 0% | 245% |

7. Figure 4 and related comments: It is hard to extract some pieces of information from this analysis because the wind at different altitudes is typically different due to the atmospheric shear layer. Additionally, different shear layers can be experienced in the same location according to the stability of the atmosphere. I strongly suggest either eliminating the plots (and the few lines of comments) from the text, or extending thoroughly the analysis as, in the current state, the plots does not offer a valuable output given the scope of the paper.

The following text has been added to the manuscript in Section 3.1 to explain Figure 4 to match the reviewer's concerns.

[revised manuscript text omitted]

9. Figure 8: missing x-label. Y-label can be improved.
Labels have been included for the "X" axis of the graph in Figure 8.

10. Pag.17, lines 254-255: As it was previously written, Here the overspeed is due to the shear layer. Since we might experience shear layers of different magnitudes, the difference between measurements taken at 10m and 40m cannot be compared directly. Hence, fig 10 and the related comments hardly provide useful information. Please, clarify.

The following text has been added to the manuscript in Section 3.2.2 to explain Figure 10 to match the reviewer's concerns.

"The representativeness of LIDAR measurements during precipitation events was assessed to determine whether the observations were consistent or affected by precipitation. Additional considerations were necessary. Ideally, an anemometer should be installed near the LIDAR at a comparable height within its vertical profile. Since this setup was not feasible, the following approach was adopted: (1) the LIDAR was installed in a flat area free of obstacles; (2) a correlation was assumed between the LIDAR's lowest range (40 m) and a sonic anemometer at 10 m on a nearby micrometeorological tower; and (3) similar correlations under precipitation and non-precipitation conditions were used as evidence that precipitation did not degrade LIDAR data quality.

This analysis involved comparing the wind speed measured by the highest anemometer (10 m) on a micrometeorological mast with the wind speed observed by the LIDAR at its lowest range (40 m). The analysis considered days with and without precipitation events. Table 16 presents the number of events analyzed, while Figure 10 illustrates the distribution of these events along with the Pearson correlation.

Analysis of the distribution of these events, with Pearson's correlation, revealed no significant change in the correlation between events with (0.93) and without precipitation (0.90), as illustrated in Figure 10.

The analysis revealed Bias values of 1.65 for events without precipitation and 1.31 for events with precipitation, reflecting the expected difference in average wind speeds between 10 m and 40 m. These findings support the conclusion that LIDAR measurements remain reliable and representative under such conditions."

11. Line 258: "suggesting that inland winds have a significant impact on the equipment's range", this needs a justification, otherwise it could be viewed as a spurious correlation. Could it be dependent on wind events, that may occur more often in the case of inland winds?

Given the reviewer's considerations that the correlation of RA drop may not be directly correlated with the wind coming from the continent, an analysis was performed with a point close to the coast (P3). The point where the most intense rainy period occurred in the entire experiment (9.78 mm/day) was chosen. The rainy period results in a decrease in the

influence of the trade winds and the intensity of the sea breeze. These two conditions lead to weaker winds and longer periods in directions coming from the continent. This means that point P3 has less influence from maritime aerosols, similar to points far from the coast. Eighteen rain events were found when analyzing this point, ten of which had RA drops. Of these, seven had speeds below 5 m/s, three had speeds between 7 and 5 m/s, seven had direction coming from the sea, and three had direction coming from the land. There were also eight precipitation events without RA drops. In all of them, the wind direction came from the sea, and the speed was above 5 m/s.

This analysis reinforces the understanding that low-intensity wind or wind from the continent delays the return of aerosols responsible for recovering RA performance.

---

## Author Comment (AC3)

Ref.: Article WES-2024-132 "Analyzing the performance of vertical wind profilers in rain events"
Author(s): Adriel J. Carvalho et al.
MS type: Research article"

Dear Reviewer2,

We have attached the comments regarding the paper "Analyzing the Performance of Vertical Wind Profilers in Rain Events" by Adriel J. Carvalho et al. The answers are in red, and the modifications are indicated in red in the manuscript.

Best wishes,

Denisson Q. Oliveira
Federal University of Maranhão, Brazil

ANSWERS TO REVIEWER2

The authors appreciate the reviewer's comments. We hope that the authors' answers and the manuscript's modifications match all the concerns. All modifications are indicated below and inserted in red in the manuscript.

The paper investigates the performance of SODAR and LIDAR vertical wind profilers during precipitation events, focusing on their Range Availability (RA) and the reliability of wind speed measurements. The study analyzes data from the Brazilian equatorial coastal region, comparing observations near and far from the shoreline. Results indicate that rainfall significantly impacts SODAR performance while having minimal effects on LIDAR near the coast. The findings aim to guide the selection of wind profiling technologies for regions with high rainfall and varying meteorological conditions. The paper has multiple unique strengths including (1) covering both SODAR and LIDAR profilers, providing a comparative assessment of their performance under varying conditions (with metrics such as Pearson Correlation); (2) addressing a critical issue for wind energy project developers in regions with high rainfall, offering actionable insights for technology selection; (3) relatively good 14 months of data (both dry and rainy) and (4) highlighting the importance of location (near vs. far from the coast) in determining the performance of LIDAR and SODAR technologies. However, multiple areas need to be addressed:

The findings are specific to the studied region, with limited discussion on how the results might apply to other geographic areas or climates.

The authors thank the reviewer for the valuable comments. The study developed in this article aims to demonstrate quantitative comparisons between the two technologies without the intention of speculating results in different climates or geographic conditions due to numerous variables that can affect the performance of the sensors that could not be quantified, such as high roughness length, low humidity, and temperature. However, on Brazil's equatorial margin, there is a large area with great wind potential and characteristics similar to those observed in this study. Section 2, "Methodology," references the papers of Assireu et al., 2022, and Pimenta et al., 2023. Both provide a detailed description of the study region, which will allow the reader to identify whether the conclusions of this work can be extrapolated to their area of interest. Additional comments are inserted into sections 2, 3, and 4.

The paper does not address the potential trade-offs between accuracy and computational or operational costs of using LIDAR versus SODAR.

This study refrained from comparing costs, transportability, and sensor software, given that there are changes over time in the prices of the various technologies, maintenance costs, component lifetimes, sensor size, and embedded software that depend on each vendor. Several vendors are on the market, and only two were used in the experiments. Such comparisons could lead the analysis to a commercial issue. The intent was to analyze the performance of the two technologies in terms of field-proven range and accuracy under

certain conditions. Additional comments regarding these points are inserted in the "Introduction".

**While Pearson correlation is used to validate the representativeness of the measurements, no detailed analysis is provided for extreme weather events or edge cases.**

The authors appreciate the suggestion. To address the reviewer's question, for the SODAR, the different rainfall intensities (C10) were compared with the consistency of rainfall (CON10). The drop in RA was found to be more related to CON10 than to C10 since, with high CON10; the RA reduction occurs for all C10 levels. As CON10 decreases, there is a smaller reduction in RA, although higher C10 values have some influence on RA for CON10 values lower than 10. A table showing RA values (mean, standard deviation, and coefficient of variation) with various C10 intensities and the maximum CON10 value was added and discussed. Additionally, for the LIDAR, it was observed that even during extreme rainfall events, as depicted in Figure 5, its performance was not affected at the time of the event, further demonstrating its resilience under high-intensity precipitation conditions. These comments have been inserted in Section 3.2.

**Key terms such as "dynamic recovery" for SODAR and specific RA thresholds for decision-making are not well-defined.**

Thanks for the suggestion. The paragraph in item 3.1 describing the recovery of the Sodar range after a precipitation event was changed to address the reviewer's observation of defining the time and range values that characterize a fast recovery. The following text has been included:

During precipitation events, the SODAR's loss of range showed a correlation with CON10 and C10. However, the pattern observed was the fast recovery of range after the precipitation ended. In other words, measurements taken after the end of precipitation generally return to 80% of the full range in the first sampling after the end of precipitation. Figure 1 (Figure 2 in the revised manuscript) depicts the equipment's range during operation when several precipitation events occurred, demonstrating the analyzed correlation.

**Some sections, such as the methodology and results, lack clarity in distinguishing novel contributions from existing literature.**

Thanks for the suggestion. The authors have updated some sections with additional comments to improve the manuscript. The contributions have been inserted in the Section 1 Introduction, and the literature review in Section 1.1 has been updated to indicate the contributions clearly.